# The Fabrication of Indium–Gallium–Zinc Oxide Sputtering Targets with Various Gallium Contents and Their Applications to Top-Gate Thin-Film Transistors

Tsung-Cheng Tien [ID], Jyun-Sheng Wu, Tsung-Eong Hsieh *[ID] and Hsin-Jay Wu

Department of Materials Science and Engineering, National Yang Ming Chiao Tung University, Hsinchu 30010, Taiwan
* Correspondence: tehsieh@nycu.edu.tw; Tel.: +886-3-5712121 (ext. 55306)

**Abstract:** We prepared amorphous indium–gallium–zinc oxide (*a*-IGZO) thin films with various Ga content ratios and investigated their feasibility as the active channel layers of top-gate thin-film transistors (TFT). First, the 2-inch IGZO sputtering targets with stoichiometric ratios of $InGaZn_2O_5$, $InGaZnO_4$, and $InGa_2ZnO_{5.5}$ were fabricated using $In_2O_3$, $Ga_2O_3$, and ZnO oxide powders as raw materials via sintering treatments at temperatures ranging from 900 °C to 1300 °C for 6 h or 8 h. *X*-ray diffraction analysis indicated that the $InGaZn_2O_5$ and $InGaZnO_4$ targets are single-phase structures whereas the $InGa_2ZnO_{5.5}$ target is a two-phase structure. Hall effect measurement indicated that the *a*-$InGaZn_2O_5$ and *a*-$InGaZnO_4$ layers possess a carrier concentration ($N$) of about $10^{19}$ cm$^{-3}$ and a resistivity ($\rho$) of about $10^{-2}$ Ω·cm; however, the $N$ of the *a*-$InGa_2ZnO_{5.5}$ layer is only $10^{17}$ cm$^{-3}$, and the $\rho$ is about 1 to 4 Ω·cm. Moreover, the *a*-$InGaZn_2O_5$ layer exhibited the highest Hall-effect mobility ($\mu_{Hall}$) of 21.17 cm$^2$·V$^{-1}$·sec$^{-1}$. This indicated that the impedance of Ga$^{3+}$ ions to carrier migration is the main factor affecting the electrical properties of *a*-IGZO layers. Ga content in the *a*-IGZO channel similarly affects the performance of the TFT devices prepared in this study. The annealing at 300 °C for 1 h in an ambient atmosphere was found to significantly improve the electrical properties of the TFT devices. The best performance was observed in the *a*-$InGaZnO_4$ TFT sample subjected to post-annealing at 300 °C with $V_{th}$ = −0.85 V, $\mu_{FE}$ = 8.46 cm$^2$, V$^{-1}$·sec$^{-1}$, $SS$ = 2.31, V·decade$^{-1}$, and $I_{on}/I_{off}$ = 2.9 × 10$^4$.

**Keywords:** IGZO sputtering target; Ga content; *a*-IGZO thin-film transistors

## 1. Introduction

Thin-film transistors (TFTs) as pixel switches are among the key components of liquid-crystal displays (LCDs) [1]. The active channel layers in TFTs directly affect the performance of LCDs, e.g., the color uniformity and energy consumption of devices. In recent years, amorphous oxide semiconductors (AOSs) have become popular channel layers for TFT devices [2]. Among them, amorphous indium–gallium–zinc oxide (*a*-IGZO) thin films have attracted considerable attention due to their visible-light transmittance, low processing temperature, high carrier mobility, and large-area uniformity. Therefore, the *a*-IGZO thin films as the channel layers of TFTs have been extensively studied [3,4]. Nomura et al. reported [5] the fabrication of transparent flexible *a*-IGZO TFT at room temperature and found that the field-effect mobility ($\mu_{FE}$) is as good as 6–9 cm$^2$·V$^{-1}$·sec$^{-1}$ after the bending of the device sample. Such a performance is much better than that of *a*-Si:H TFTs [6], and the process cost is lower than that of poly-Si TFTs.

IGZO is composed of three metal oxides: indium oxide ($In_2O_3$) [7], gallium oxide ($Ga_2O_3$) [8], and zinc oxide (ZnO) [9], which have distinct effects on the physical properties of IGZO layers. $In_2O_3$ mainly contributes to mobility due to having larger *n*s orbitals of metal cations than the 2*p* orbitals of oxygen anions [10]. Nevertheless, $In_2O_3$ and ZnO are inherent oxygen-deficiency oxides which contain a large number of oxygen vacancies.

This results in the high carrier concentration ($N > 10^{17}$ cm$^{-3}$) of IGZO [5] which, in turn, causes TFTs in the normally-on state and induces a serious leakage current problem. To overcome this difficulty, $Ga_2O_3$ is added since the $Ga^{3+}$ ions in $Ga_2O_3$ may combine with the oxygen ions to reduce the number of oxygen vacancies and, hence, modulate the $N$ value of IGZO layers.

In this study, we investigated the influence of Ga content on the transport properties of IGZO layers and on the electrical properties of IGZO TFTs. The IGZO sputtering targets with stoichiometric ratios of $InGaZn_2O_5$, $InGaZnO_4$, and $InGa_2ZnO_{5.5}$ were prepared by using the target fabrication method established previously [11]. The phase evolution during the sintering treatment of the IGZO targets and, accordingly, the microstructures and electrical properties of the IGZO layers were analyzed. Afterward, the TFT devices containing IGZO active channel layers with various Ga contents were prepared, and their transfer characteristics including threshold voltage ($V_{th}$), field-effect mobility ($\mu_{FE}$), subthreshold swing ($SS$), and on/off current ratio ($I_{on}/I_{off}$) were investigated. The effects of post-annealing on device performance were also examined and discussed as follows.

## 2. Experimental Methods

### 2.1. Preparation of IGZO Precursor Powder Mixtures

High-quality commercial $In_2O_3$, $Ga_2O_3$, and ZnO powders (purity > 99.99%; supplier: ELECMAT/USA) were adopted as the raw materials for the fabrication of the sputtering target. First, the oxide powders were uniformly mixed at the stoichiometric ratios listed in Table 1, and the ethanol and 3 wt.% polymethacrylic acid (PMAA) serving as the chemical dispersion were added. Mechanical milling (MiniZeta 03 Laboratory Mill) at 2400 rpm for 15 min was then performed to obtain the IGZO precursor powder mixtures in nanometer scale. We note that the high specific surface-area feature of nanoscale precursor powders benefits the phase formation in IGZO targets during subsequent sintering treatment [12].

**Table 1.** Stoichiometry and Ga contents of IGZO targets.

| Stoichiometry of IGZO Targets | Ga (*at.*%) |
| :---: | :---: |
| $InGaZn_2O_5$ | 25 |
| $InGaZnO_4$ | 33.33 |
| $InGa_2ZnO_{5.5}$ | 50 |

### 2.2. Preparation and Characterization of IGZO Targets

The IGZO precursor powder mixtures were dried in a furnace, sifted in a 200-mesh sieve, poured into a 2-inch model, and formed into the target green bodies at the pressure of 100 MPa. Then pressure-less sintering at 900 °C, 1100 °C, 1200 °C, and 1300 °C for 6 to 8 h was performed to obtain IGZO sputtering targets. In the meantime, x-ray diffraction (XRD, Bruker Instruments D2 Phaser) with Cu-$K_{\alpha}$ radiation ($\lambda$ = 0.154 nm) and scan angle ($2\theta$) ranging from 10° to 70° at a scan rate of 0.05° sec$^{-1}$ was performed to analyze the phase evolution of the IGZO targets sintered at various temperatures. The purpose of the XRD analysis was to determine the correlations between crystalline phase and sintering conditions so as to establish the optimal sintering treatment of the IGZO sputtering targets.

Moreover, the Archimedes method [13] using an electronic balance was adopted to measure the relative densities of the IGZO targets. Previous studies reported that the theoretical density of an IGZO target is 6.38 g·cm$^{-3}$ [14,15] and, via measurement and calculation, the relative densities of the IGZO targets prepared in this study were found to be about 90%; in other words, the densities of the IGZO targets are about 5.74 g·cm$^{-3}$.

### 2.3. Preparation and Characterization of IGZO TFT Device Samples

Figure 1a,b sketches the cross-sectional and top-view device structures of the top-gate *a*-IGZO TFT devices prepared in this study. The devices have a channel length ($L$) of 10 μm and a channel width ($W$) of 100 μm, as shown in Figure 1b. First the *n*-type Si

wafer substrates were cleaned using the RCA method and 300-nm-thick silicon dioxide ($SiO_2$) buffer layers were grown on the Si substrates using the wet oxidation process. Then 25-nm-thick *a*-IGZO channel layers were deposited using the radio frequency (RF) magnetron sputtering method in a self-assembled six-target sputtering system at a background pressure of $3 \times 10^{-6}$ torr. The deposition was performed at room temperature with the conditions of: working pressure = 3 mtorr, RF power = 80 W, Ar gas flow rate = 20 sccm, and $O_2$ gas flow = 0 sccm or 1.2 sccm. Afterward, 100-nm-thick molybdenum (Mo) layers as the source/drain (S/D) electrodes of devices were deposited by RF sputtering at room temperature with the conditions of working pressure = 3 mtorr, RF power = 100 W, and Ar gas flow rate = 10 sccm. Note that the S/D electrode patterns were accomplished by the photolithography process in conjunction with lift-off techniques. Sequentially, a gate dielectric oxide layer comprised of a stack of $SiN_y$ (thickness = 35 nm) and $SiO_x$ (thickness = 165 nm) was deposited on the active channel region by plasma-enhanced chemical vapor deposition (PECVD, OXFORD Instruments Plasmalab 80Plus, Oxford Instruments, Abingdon-on-Thames, UK). After the gate dielectric oxide deposition, part of the samples was annealed at 300 °C for 1 h in an ambient atmosphere. Finally, the Mo gate electrode was deposited on the gate dielectric oxide with the same conditions as the those in the preparation of the S/D electrodes. A post-annealing at 300 °C for 1 h in an ambient atmosphere was also performed for part of the TFT samples so as to explore the influence of heat treatment on the device performance.

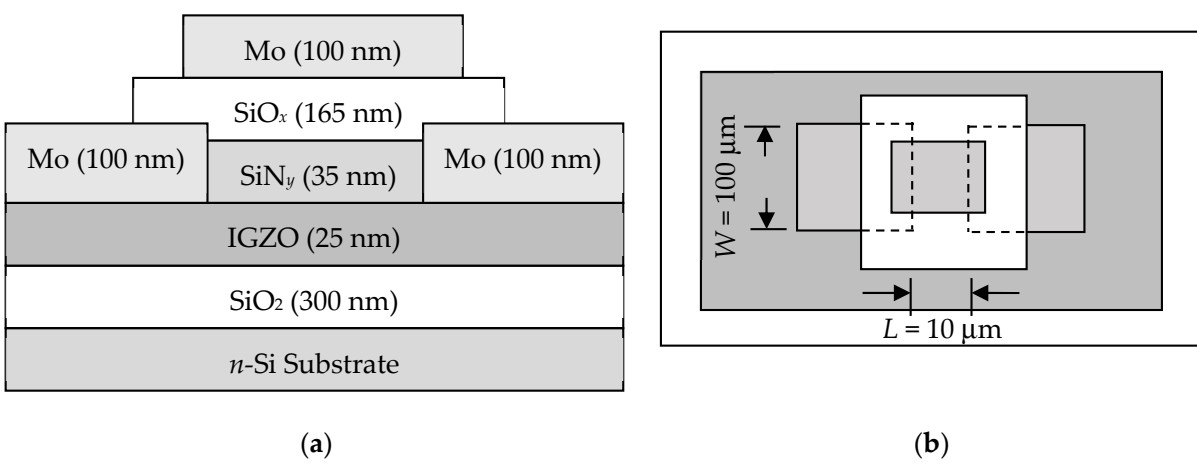

(**a**)   (**b**)

**Figure 1.** (**a**) Cross-sectional and (**b**) top-view device structures of the *a*-IGZO TFT samples.

Scanning electron microscopy (SEM, JEOL 6500) with an energy-dispersive spectrometer (EDS, Oxford Inca Energy 300, Oxford Instruments) was adopted to analyze the morphology and element distribution of the IGZO layers. SEM was also adopted to characterize the layer thickness of the TFT samples. The phase structure of the IGZO layers was characterized by XRD (Bruker Instruments D8 Discover, Bruker Billerica, MA, USA) with Cu-$K_\alpha$ radiation in $2\theta$ ranging from 10° to 70° at a scan rate of 0.03° sec$^{-1}$. The electrical properties (*N*, Hall mobility ($\mu_{Hall}$), resistivity ($\rho$)), and optical properties of the IGZO layers were determined by a Hall effect measurement system (Ecopia HMS-3000) and an UV-NIR spectrometer (JASCO V-670), respectively. Note that the IGZO layers deposited on glass substrates are the samples for transmittance measurement. The transfer characteristics of the IGZO TFT devices were measured using a precision semiconductor parameter analyzer (Keysight B1500A, Keysight, Santa Rosa, CA, USA). To fulfill the calculation of the transfer characteristics of the IGZO TFTs, a metal–insulator–metal (MIM) structure containing an $SiO_x$/$SiN_y$ gate dielectric layer was also fabricated and its capacitance was measured by a precision LCR meter (Wayne Kerr 6520B, Islamabad, Pakistan) at 1 MH$_Z$.

## 3. Results and Discussions

### 3.1. Characterization of the IGZO Sputtering Targets

Figure 2a–c presents the XRD profiles of the IGZO targets with various Ga content ratios sintered in temperatures of 900 °C to 1300 °C for 6 or 8 h. According to Joint Committee of Powder Diffraction Standard (JCPDS) files nos. 400,252 and 381,104, both the $InGaZn_2O_5$ and the $InGaZnO_4$ targets are of single-phase structure when the sintering temperature reaches 1300 °C, as shown in Figure 2a,b. However, the XRD pattern of the $InGa_2ZnO_{5.5}$ target shown in Figure 2c indicates that it contains two phases, i.e., $Ga_2ZnO_4$ and $In_2O_3$, in accord with the JCPDS files nos. 860,410 and 060,416. Hence, the $InGa_2ZnO_{5.5}$ target sintered at the temperature of 1300 °C is of two-phase structure.

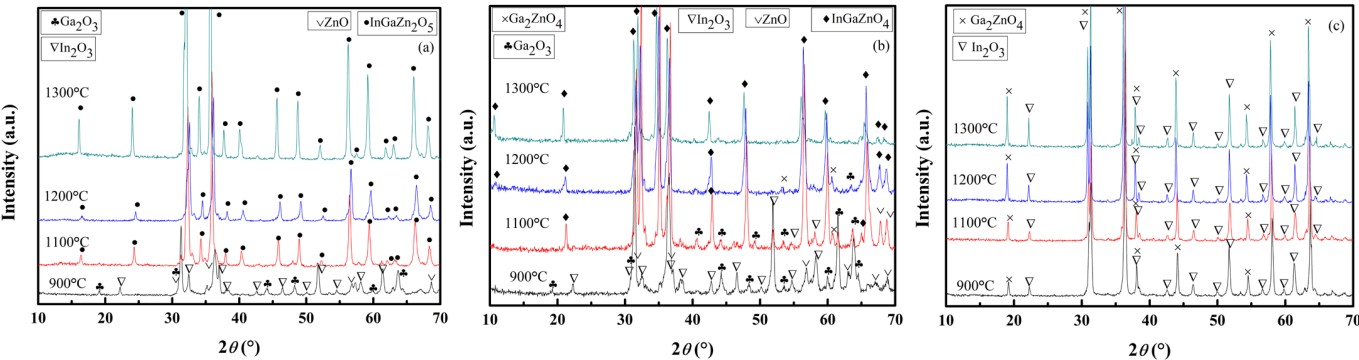

**Figure 2.** XRD patterns of (**a**) $InGaZn_2O_5$, (**b**) $InGaZnO_4$, and (**c**) $InGa_2ZnO_{5.5}$ sputtering targets sintered at temperatures from 900 °C to 1300 °C for 6 or 8 h.

Figure 2a,b illustrates that the IGZO targets remain as the mixtures of raw powders at a temperature of 900 °C. According to Figure 2a, the $InGaZn_2O_5$ phase formed in $InGaZn_2O_5$ target at 1100 °C via the following reaction:

$$In_2O_3 + Ga_2O_3 + 4ZnO \rightarrow 2InGaZn_2O_5 \ (1100 \ °C) \tag{1}$$

Figure 2b shows that at 1100 °C the cubic spinel $Ga_2ZnO_4$ and the rhombohedral $InGaZnO_4$ phases emerged in the $InGaZnO_4$ target according to the following reactions:

$$Ga_2O_3 + ZnO \rightarrow Ga_2ZnO_4 \ (1100 \ °C) \tag{2}$$

$$In_2O_3 + Ga_2ZnO_4 + ZnO \rightarrow 2InGaZnO_4 \ (1100 \ °C) \tag{3}$$

With the increase in sintering temperature, the $Ga_2ZnO_4$ reacted with the remaining raw powders to form the $InGaZnO_4$ phase, and the $InGaZnO_4$ target became the single-phase structure at 1300 °C.

For the $InGa_2ZnO_{5.5}$ target, the $Ga_2O_3$ and ZnO reacted to form $Ga_2ZnO_4$ phases at 900 °C. $In_2O_3$ acted as an inert component during the sintering treatment of the $InGa_2ZnO_{5.5}$ target up to 1300 °C and, hence, the two phases, $In_2O_3$ and $Ga_2ZnO_4$, comprised the $InGa_2ZnO_{5.5}$ target.

### 3.2. Characterization of a-IGZO Channel Layers

The IGZO layers were deposited using self-prepared IGZO targets with various Ga content ratios. Figure 3a,b shows the XRD patterns of the as-deposited IGZO layers and the IGZO layers subjected to annealing at 300 °C for 1 h in an ambient atmosphere. Regardless of whether annealing treatment was carried out or not, the characteristic diffraction peaks were absent for the three types of IGZO layers in the $2\theta$ range of 10° to 70°. The analytical results indicate that the phase structures of the IGZO sputtering targets and the post-annealing have no effect on the amorphism of the IGZO layers prepared in this study [16].

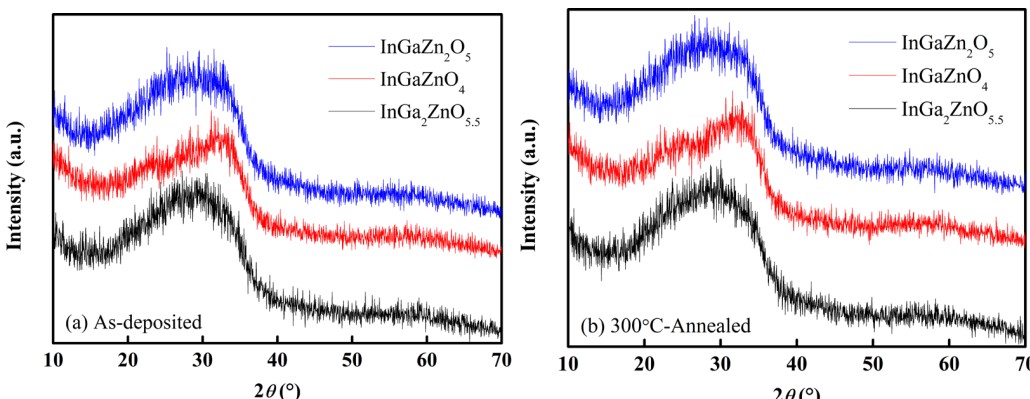

**Figure 3.** XRD patterns of (**a**) as-deposited and (**b**) 300 °C-annealed IGZO layers prepared by using IGZO targets with various Ga content ratios.

Figure 4a,b shows the transmittance spectra of the *a*-IGZO layers in the visible-light wavelength range of 300 to 800 nm. The transmittances are greater than 80%, and there is no significant difference in transmittances for all the types of *a*-IGZO layers. The types and ratios of gas flow rates for the sputtering deposition also show little influence on the transmittances of the *a*-IGZO layers prepared in this study.

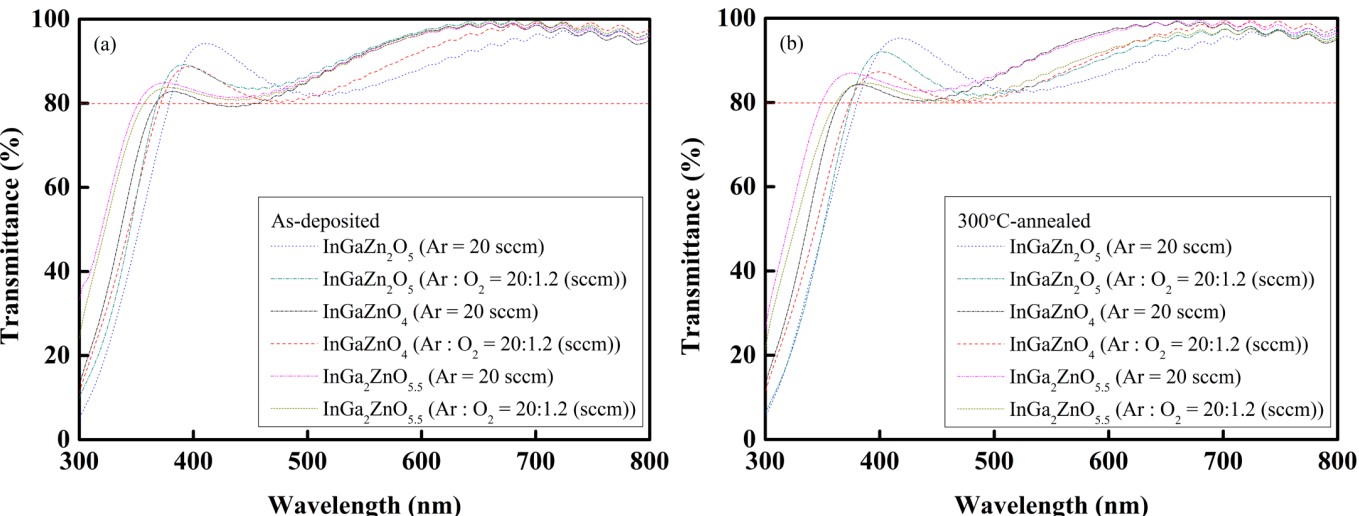

**Figure 4.** The transmittance profiles of (**a**) as-deposited and (**b**) 300 °C-annealed *a*-IGZO layers deposited with Ar = 20 sccm and Ar:$O_2$ = 20:1.2 (sccm) gas flow rates.

Figure 4a,b also reveals the shift in the transmittance spectra of the *a*-IGZO layers, and this was found to relate to the Ga content of the *a*-IGZO layers. Figure 5a,b sketches the $(\alpha h\nu)^2$ versus $h\nu$ plots extracted from the transmittance spectra, and the Tauc's method [17] was employed to determine the values of the energy bandgap ($E_g$'s) of the *a*-IGZO layers. The analytical results indicated that the $E_g$ value increased with the increase in the Ga content of the *a*-IGZO layers. For instance, in the 300 °C-annealed samples deposited without oxygen incorporation, the $E_g$'s of the *a*-InGaZn$_2$O$_5$, *a*-InGaZnO$_4$, and *a*-InGa$_2$ZnO$_{5.5}$ layers were separately equal to 3.73, 3.98, and 4.03 eV. This is ascribable to the strong bonding feature of the Ga$^{3+}$ ions which, in turn, limits the motion of charge carriers in the *a*-IGZO layers. In other words, the enlargement of the $E_g$ value of an *a*-IGZO layer due to the increase in Ga content predicts that the charge carriers will be more difficult to migrate in an *a*-IGZO layer with high Ga content.

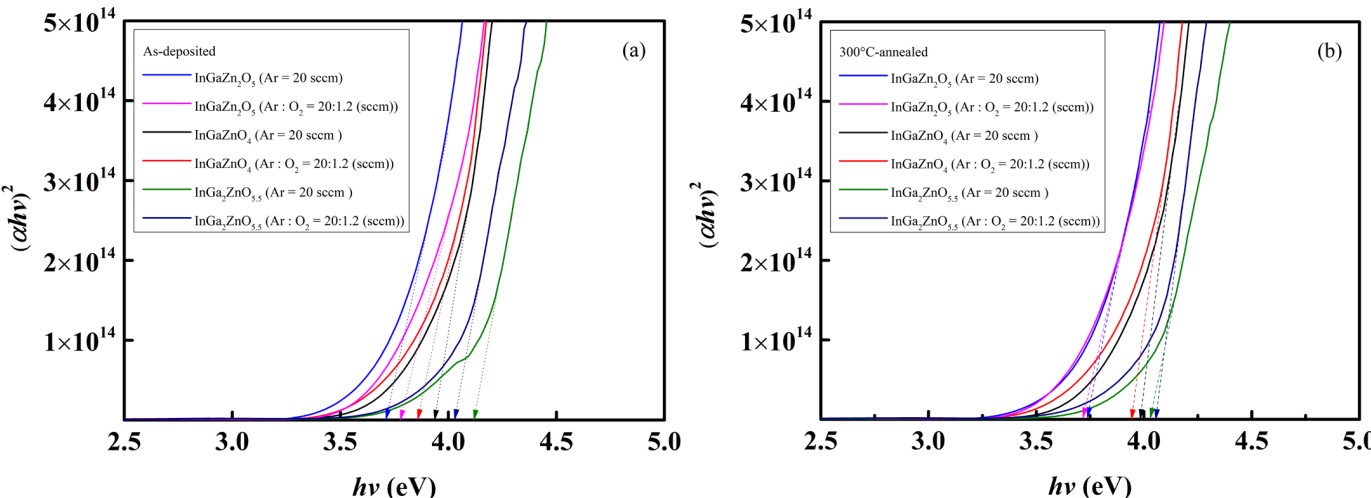

**Figure 5.** The $(\alpha h\nu)^2$ versus $h\nu$ plots of (**a**) as-deposited and (**b**) 300 °C-annealed *a*-IGZO layers deposited with Ar = 20 sccm and Ar:O$_2$ = 20:1.2 (sccm) gas flow rates.

Figure 6a–c depicts the *N*, $\mu_{Hall}$, and $\rho$ of various *a*-IGZO layers deposited with the gas flow rates of Ar = 20 sccm, and Ar:O$_2$ = 20:1.2 (sccm), and their values are listed in Tables 2 and 3. The Hall effect measurement revealed that all the *a*-IGZO layers are of the *n*-type and, with the increase in Ga content, the *N* and $\mu_{Hall}$ decrease whereas the $\rho$ increases. Hosono has reported that, due to the relatively small ionic radius, the Ga$^{3+}$ ions may attract oxygen ions in *a*-IGZO layers and thereby suppress the formation of oxygen vacancies [18]. The oxygen-deficiency reaction in ionic compounds is known to be:

$$O_O \rightarrow 2e' + V_O^{\bullet\bullet} + \frac{1}{2}O_{2(g)} \tag{4}$$

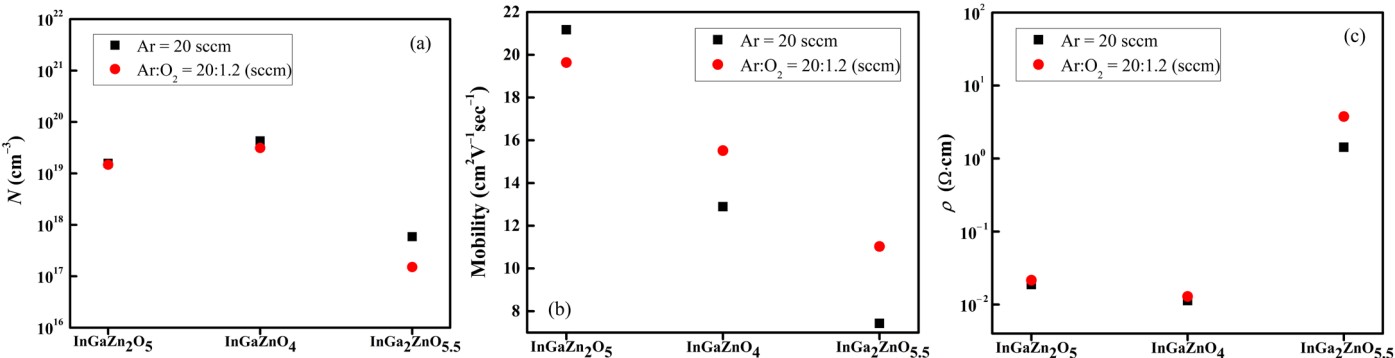

**Figure 6.** Hall effect measurement of (**a**) *N*, (**b**) $\mu_{Hall}$, and (**c**) $\rho$ of the various *a*-IGZO thin films with Ar = 20 sccm and Ar:O$_2$ = 20:1.2 (sccm).

**Table 2.** Hall measurement result of the various *a*-IGZO thin films with Ar = 20 sccm.

| *a*-IGZO Layer Type | *N* (cm$^{-3}$) | $\mu_{Hall}$ (cm$^2\cdot$V$^{-1}\cdot$sec$^{-1}$) | $\rho$ ($\Omega\cdot$cm) |
|---|---|---|---|
| *a*-InGaZn$_2$O$_5$ | $1.57 \times 10^{19}$ | 21.17 | $1.88 \times 10^{-2}$ |
| *a*-InGaZnO$_4$ | $4.29 \times 10^{19}$ | 12.89 | $1.13 \times 10^{-2}$ |
| *a*-InGa$_2$ZnO$_{5.5}$ | $5.88 \times 10^{17}$ | 7.43 | 1.43 |

**Table 3.** Hall measurement result of the various $a$-IGZO thin films with Ar:$O_2$ = 20:1.2 (sccm).

| $a$-IGZO Layer Type | $N$ (cm$^{-3}$) | $\mu_{\text{Hall}}$ (cm$^2 \cdot$V$^{-1} \cdot$sec$^{-1}$) | $\rho$ ($\Omega \cdot$cm) |
|---|---|---|---|
| $a$-InGaZn$_2$O$_5$ | $1.48 \times 10^{19}$ | 19.64 | $2.15 \times 10^{-2}$ |
| $a$-InGaZnO$_4$ | $3.12 \times 10^{19}$ | 15.52 | $1.29 \times 10^{-2}$ |
| $a$-InGa$_2$ZnO$_{5.5}$ | $1.51 \times 10^{17}$ | 11.03 | 3.76 |

According to the above reaction, the number of electrons decreases with the decrease in oxygen vacancies and thereby the $N$ decreases with the increase in Ga content in $a$-IGZO layers. This is clearly supported by the measured results that the $N$'s of the $a$-InGaZn$_2$O$_5$ and $a$-InGaZnO$_4$ layers are about $10^{19}$ cm$^{-3}$ whereas the $N$ of the $a$-InGa$_2$ZnO$_{5.5}$ layer decreases by two orders of magnitude to $10^{17}$ cm$^{-3}$. The relatively high valence feature of the Ga$^{3+}$ ions may also attract electrons and impede their migration, which explains why the $\mu_{Hall}$ decreases with the increase in Ga content in the $a$-IGZO layers. The electrical conductivity ($\sigma$) can be expressed as:

$$\sigma = \frac{1}{\rho} \propto Ne\mu_{Hall} \tag{5}$$

where $\rho$ is the electrical resistivity and $e$ is the electronic charge. As indicated by Equation (5), the decrease in $N$ and $\mu_{Hall}$ leads to the increase in $\rho$. The results of the Hall effect measurement are hence consistent with the results reported by Hosono [18].

In this study, oxygen gas was also introduced during the sputtering deposition of the $a$-IGZO layers, and its influence on electrical properties of the $a$-IGZO layers was investigated. Figure 6a shows that the $N$'s of the $a$-IGZO layers deposited without the introduction of oxygen gas are slightly higher than those of the $a$-IGZO layers deposited with the introduction of oxygen gas. This is ascribed to the remedy of oxygen deficiencies in the $a$-IGZO layers deposited with the introduction of the oxygen gas, which consequently reduced the numbers of oxygen vacancies and charge carriers as depicted by Equation (4). The reduction in charge carriers also caused the escalation of $\rho$ in the $a$-IGZO layers deposited with the introduction of oxygen gas, as shown in Figure 6c. As to the $\mu_{\text{Hall}}$, Figure 6b indicates that the values of $\mu_{\text{Hall}}$ decreased with the increase in Ga content in the $a$-IGZO layers regardless of the introduction of the oxygen gas during the sputtering deposition. As revealed by the XRD analysis, all the IGZO layers prepared in this study are amorphous, so that the lattice regularity should have little influence on the $\mu_{\text{Hall}}$ values. It is hence speculated that the difference in Ga content dominates the mobility property of the $a$-IGZO layers prepared in this study. As stated above, the high valence feature of the Ga$^{3+}$ ions might impede the migration of the charge carriers. The $a$-InGaZn$_2$O$_5$ layers possesses the lowest Ga content, as listed in Table 1, implying less obstacles of charge carrier migration induced by Ga$^{3+}$ ions and, hence, a high $\mu_{\text{Hall}}$ value for the $a$-InGaZn$_2$O$_5$ layer. Moreover, the $a$-InGaZn$_2$O$_5$ layer deposited within Ar = 20 sccm possesses the highest $\mu_{\text{Hall}}$ of 21.17 cm$^2 \cdot$V$^{-1} \cdot$sec$^{-1}$, which is higher than that of the $a$-InGaZn$_2$O$_5$ layers deposited with Ar:$O_2$ = 20:1.2 (sccm) as depicted by Figure 6b. On the contrary, the sputtering deposition with Ar:$O_2$ = 20:1.2 (sccm) benefits the $\mu_{\text{Hall}}$'s of the $a$-InGaZnO$_4$ and $a$-InGa$_2$ZnO$_{5.5}$ layers as shown in Figure 6b. We speculated that the remedy for oxygen deficiencies caused by the introduction of the oxygen gas during the sputtering deposition similarly repairs the interruptions of atom bonds caused by the oxygen vacancies and improves the $\mu_{\text{Hall}}$'s of the $a$-InGaZnO$_4$ and $a$-InGa$_2$ZnO$_{5.5}$ layers.

Figure 7a–f presents the SEM micrographs and EDS element mappings of as-deposited and 300 °C-annealed $a$-IGZO layers deposited with a gas flow rate of Ar:$O_2$ = 20:1.2 (sccm). The analytical results of SEM and EDS indicate that all the $a$-IGZO layers exhibit flat surfaces with no specific geometric features and uniform In, Ga, and Zn element distributions. As depicted by Figure 7e,f, element aggregation was absent even in the $a$-InGa$_2$ZnO$_{5.5}$ layer prepared by using the double-phase sputtering target. This illustrates that Ga content is indeed the main factor affecting the electrical properties of $a$-IGZO layers.

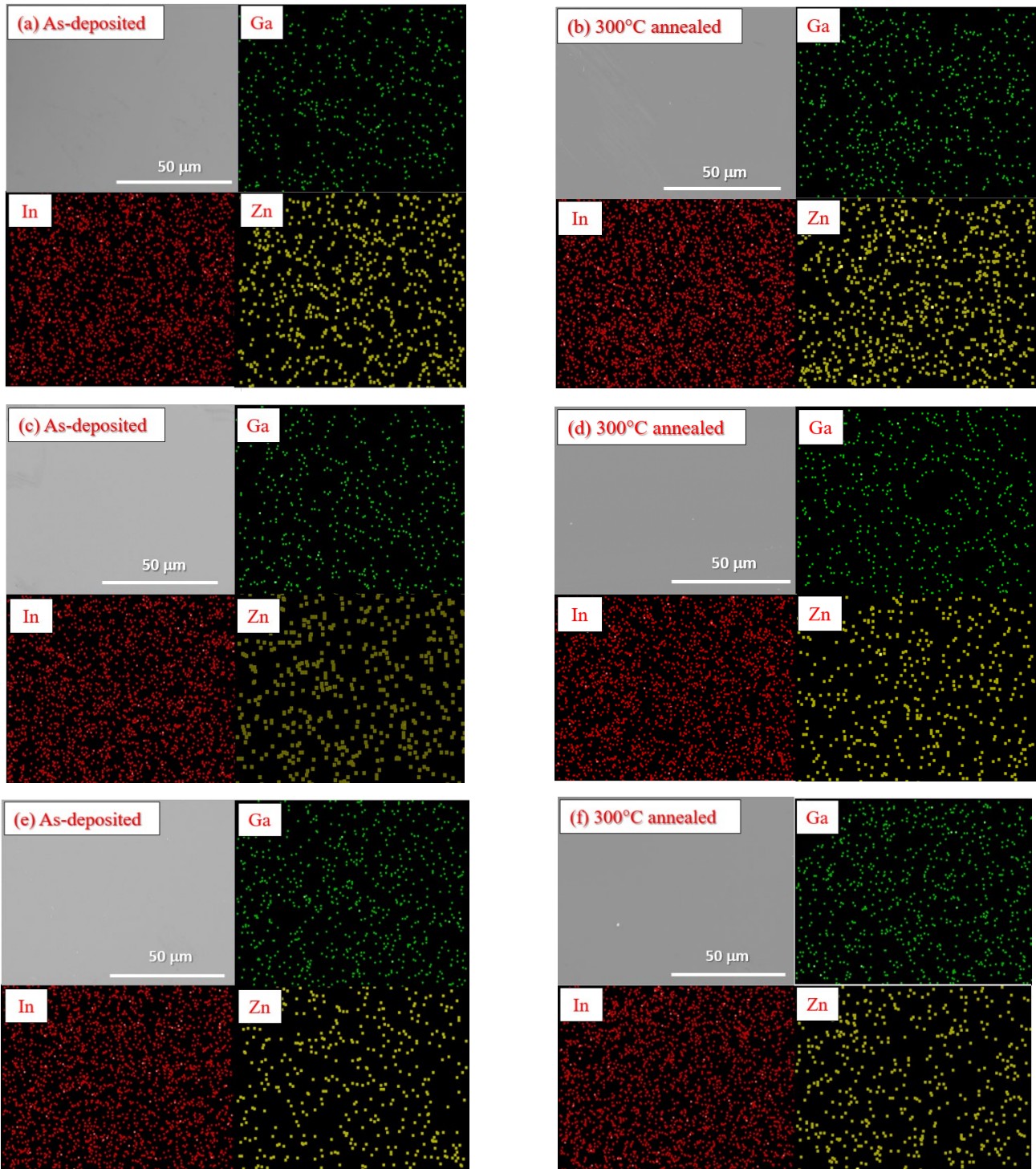

**Figure 7.** SEM micrographs and EDS element mappings of the three types of *a*-IGZO layers deposited with Ar:O$_2$ = 20:1.2 (sccm). (**a**) as-deposited and (**b**) 300 °C-annealed *a*-InGaZn$_2$O$_5$ layers; (**c**) as-deposited and (**d**) 300 °C-annealed *a*-InGaZnO$_4$ layers; (**e**) as-deposited and (**f**) 300 °C-annealed *a*-InGa$_2$ZnO$_{5.5}$ layers.

### 3.3. Electrical Properties of the Three Types of IGZO TFT Devices

In the TFT samples prepared by this study, the gate dielectric material was a stack of SiO$_x$ and SiN$_y$ with the dielectric constants of 3.9 and 7.5 [19], respectively. For subsequent calculations of transfer characteristics of *a*-IGZO TFTs, we fabricated MIM samples with Mo/SiO$_x$/SiN$_y$/Mo structure and measured their capacitance-versus-voltage (*C-V*) profiles

at 1 MHz. As shown in Figure 8, one can see that the total capacitances of as-deposited and 300 °C-annealed samples are nearly the same.

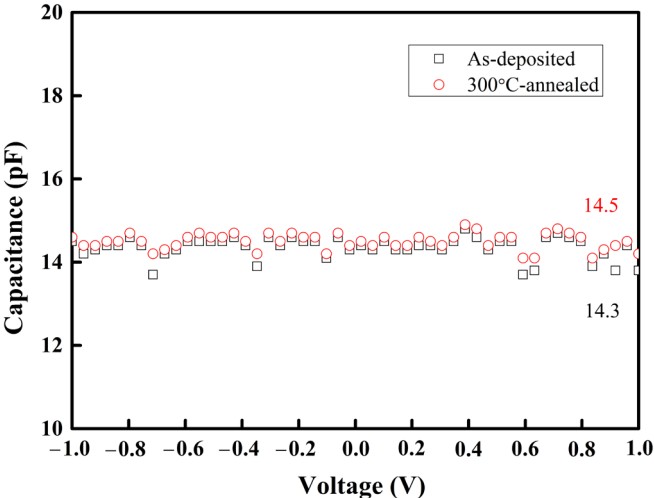

**Figure 8.** *C-V* profiles of as-deposited and 300 °C-annealed Mo/SiO$_x$/SiN$_y$/Mo MIM samples.

Figures 9a–c and 10a–c present separately the transfer and output characteristics of the TFT samples containing the three types of as-deposited *a*-IGZO active channel layers. The transfer characteristics, i.e., $V_{th}$, $\mu_{FE}$, *SS*, and $I_{on}/I_{off}$, were calculated in accordance with the data presented in Figure 9a–c, and the results are summarized in Table 4. The $V_{th}$'s of all the TFTs are about 2 V, indicating that a positive voltage is required for carrier extraction. Although the performance of TFT containing an *a*-InGaZnO$_4$ channel layer is the best among the three types of TFT samples, its $\mu_{FE}$ is only 1.65 cm$^2$·V$^{-1}$·sec$^{-1}$, as shown in Table 4. A previous *C-V* study performed in our laboratory [11] demonstrated a rather rapid shift from depletion region to accumulation region in annealed samples, and a reduction in trap density at an *a*-IGZO/SiO$_2$ interface for about one order of magnitude was observed. This implied that a post-annealing at 300 °C for 1 h in an ambient atmosphere may lead to a well-formed *a*-IGZO/SiO$_2$ interface in the TFT sample. This study hence carried out the annealing of a gate dielectric layer and the post-annealing of TFT samples to improve the device performance. The annealing of the gate dielectric layer was intended to modulate the Mo/IGZO interface and suppress the defects at the SiO$_x$/SiN$_y$/Mo interface, and the post-annealing was intended to eliminate the water and oxygen adsorption in the ambient atmosphere during the device fabrication and reduce the interfacial defects between the layers of TFTs [20,21].

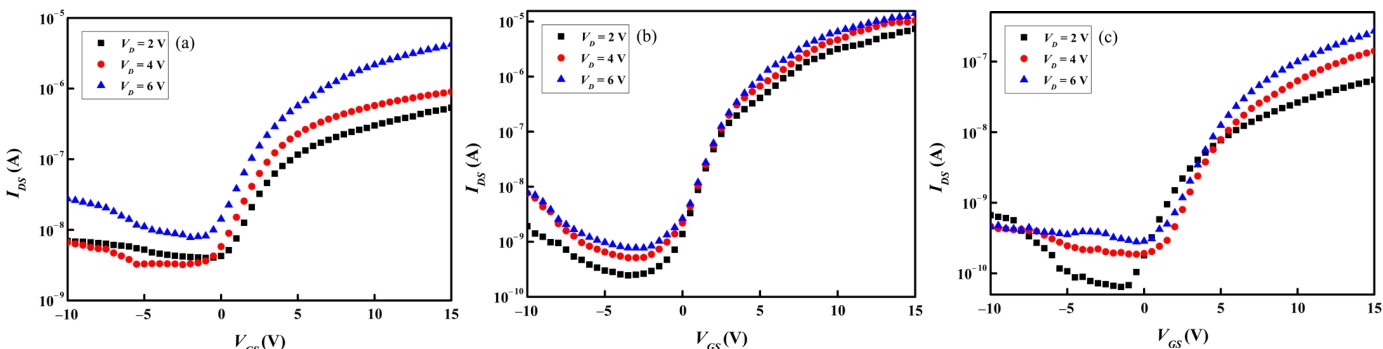

**Figure 9.** Transfer characteristics of TFT samples containing as-deposited (**a**) *a*-InGaZn$_2$O$_5$, (**b**) *a*-InGaZnO$_4$, and (**c**) *a*-InGa$_2$ZnO$_{5.5}$ active channel layers.

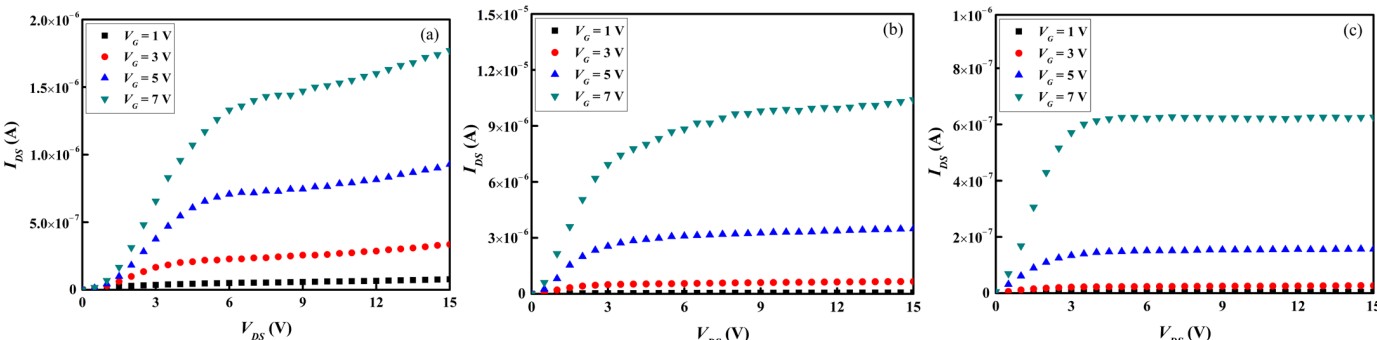

**Figure 10.** Output characteristics of TFT samples containing as-deposited (**a**) *a*-InGaZn$_2$O$_5$, (**b**) *a*-InGaZnO$_4$, and (**c**) *a*-InGa$_2$ZnO$_{5.5}$ active channel layers.

**Table 4.** Transfer characteristics of the three types of TFT samples containing as-deposited *a*-IGZO active channel layers.

| *a*-IGZO Layer Type | $V_{th}$ (V) | $\mu_{FE}$ (cm$^2$·V$^{-1}$·sec$^{-1}$) | $SS$ (V·Decade$^{-1}$) | $I_{on}/I_{off}$ |
|---|---|---|---|---|
| *a*-InGaZn$_2$O$_5$ | 1.91 | 0.48 | 2.87 | $2.1 \times 10^3$ |
| *a*-InGaZnO$_4$ | 1.87 | 1.65 | 1.37 | $5.4 \times 10^5$ |
| *a*-InGa$_2$ZnO$_{5.5}$ | 2.26 | 0.05 | 3.51 | $1.4 \times 10^3$ |

Figures 11a–c and 12a–c present separately the transfer and output characteristics of the three types of *a*-IGZO TFT devices subjected to the gate dielectric annealing at 300 °C for 1 h in an ambient atmosphere. As listed in Table 5, which summarizes the transfer characteristics of such TFT samples, one can see that the device performance is improved. Compared to the TFT samples containing as-deposited *a*-IGZO channel layers, $V_{th}$ shifts toward the negative bias and both *SS*, $\mu_{FE}$, and $I_{on}/I_{off}$ are improved.

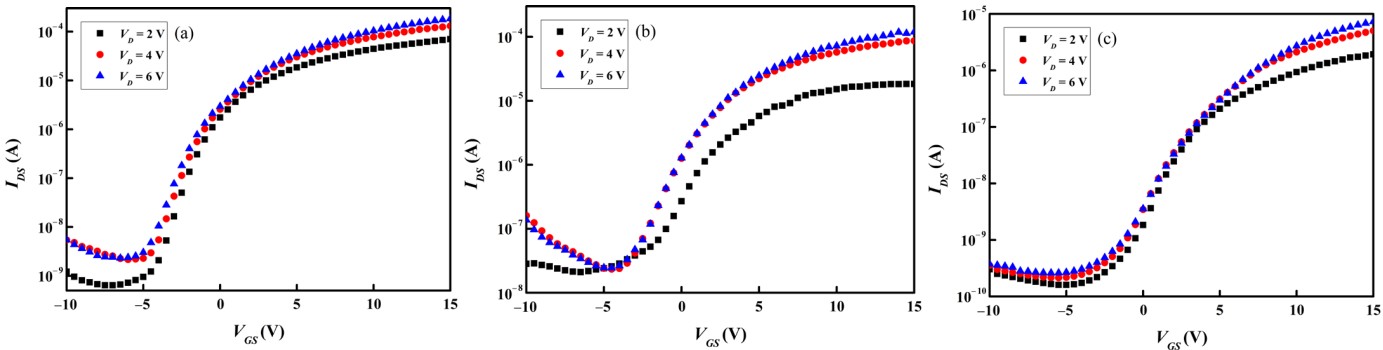

**Figure 11.** Transfer characteristics of TFT samples containing (**a**) *a*-InGaZn$_2$O$_5$, (**b**) *a*-InGaZnO$_4$, and (**c**) *a*-InGa$_2$ZnO$_{5.5}$ active channel layers subjected to the gate dielectric annealing at 300 °C for 1 h in an ambient atmosphere.

**Table 5.** Transfer characteristics of the three types of *a*-IGZO TFT samples subjected to the gate dielectric annealing at 300 °C for 1 h in an ambient atmosphere.

| *a*-IGZO Layer Type | $V_{th}$ (V) | $\mu_{FE}$ (cm$^2$·V$^{-1}$·sec$^{-1}$) | $SS$ (V·Decade$^{-1}$) | $I_{on}/I_{off}$ |
|---|---|---|---|---|
| *a*-InGaZn$_2$O$_5$ | −2.27 | 7.61 | 1.12 | $1.3 \times 10^5$ |
| *a*-InGaZnO$_4$ | −1.33 | 5.47 | 1.32 | $2.0 \times 10^4$ |
| *a*-InGa$_2$ZnO$_{5.5}$ | 0.59 | 0.48 | 2.06 | $3.6 \times 10^3$ |

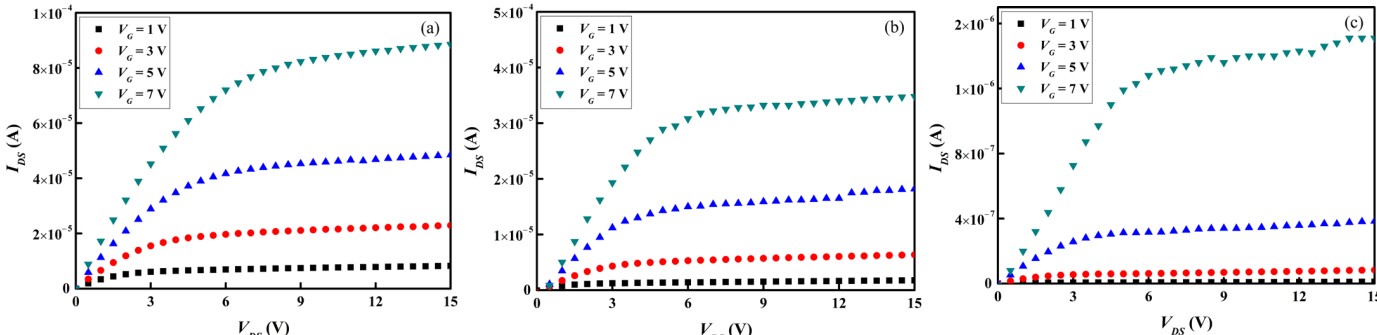

**Figure 12.** Output characteristics of TFT samples containing (**a**) *a*-InGaZn$_2$O$_5$, (**b**) *a*-InGaZnO$_4$, and (**c**) *a*-InGa$_2$ZnO$_{5.5}$ active channel layers subjected to the gate dielectric annealing at 300 °C for 1 h in an ambient atmosphere.

Figures 13a–c and 14a–c present separately the transfer and output characteristics of the three types of *a*-IGZO TFT devices subjected to post-annealing at 300 °C for 1 h in an ambient atmosphere. The transfer characteristics listed in Table 6 show that the $V_{th}$ further shifts toward the negative bias and that the $\mu_{FE}$ is significantly improved. The *a*-InGaZn$_2$O$_5$ TFT device exhibits the highest $\mu_{FE}$ of 9.95 cm$^2$·V$^{-1}$·sec$^{-1}$, whereas the $\mu_{FE}$ of the *a*-InGa$_2$ZnO$_{5.5}$ TFT device is the lowest of 1.97 cm$^2$·V$^{-1}$·sec$^{-1}$. This is possibly ascribable to the difference in Ga content in the *a*-IGZO channel layer of the TFT samples, which affects the mobility property of the *a*-IGZO layers, as stated in the previous section. Overall, the best performance was observed in the *a*-InGaZnO$_4$ TFT sample subjected to the 300 °C-post annealing with $V_{th} = -0.85$ V, $\mu_{FE} = 8.46$ cm$^2$·V$^{-1}$·sec$^{-1}$, $SS = 2.31$ V·decade$^{-1}$, and $I_{on}/I_{off} = 2.9 \times 10^4$.

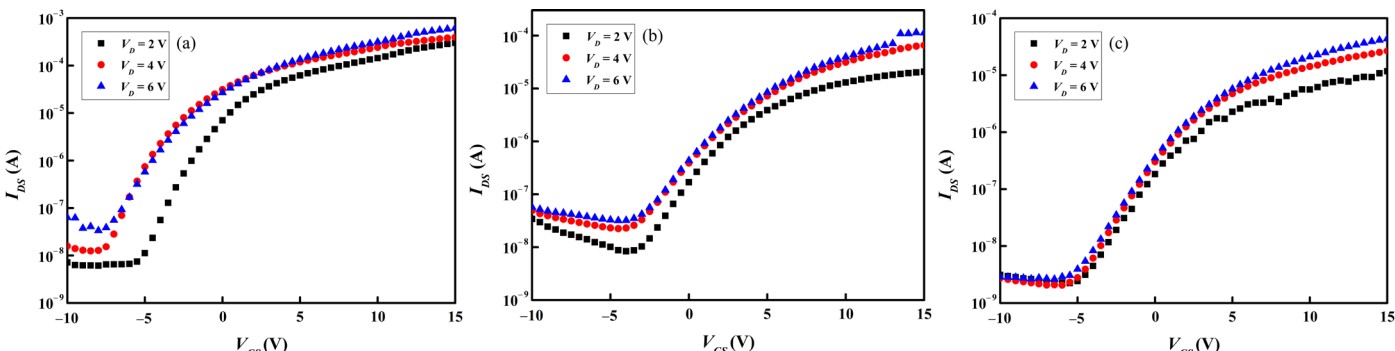

**Figure 13.** Transfer characteristics of TFT devices containing (**a**) *a*-InGaZn$_2$O$_5$, (**b**) *a*-InGaZnO$_4$, and (**c**) *a*-InGa$_2$ZnO$_{5.5}$ active channel layers subjected to post-annealing at 300 °C for 1 h in an ambient atmosphere.

**Table 6.** Transfer characteristics of the three types of *a*-IGZO TFT samples subjected to post-annealing at 300 °C for 1 h in an ambient atmosphere.

| *a*-IGZO Layer Type | $V_{th}$ (V) | $\mu_{FE}$ (cm$^2$·V$^{-1}$·sec$^{-1}$) | $SS$ (V·Decade$^{-1}$) | $I_{on}/I_{off}$ |
|---|---|---|---|---|
| *a*-InGaZn$_2$O$_5$ | −3.26 | 9.95 | 1.89 | $2.1 \times 10^4$ |
| *a*-InGaZnO$_4$ | −0.85 | 8.46 | 2.31 | $2.9 \times 10^4$ |
| *a*-InGa$_2$ZnO$_{5.5}$ | −1.25 | 1.97 | 2.36 | $7.9 \times 10^3$ |

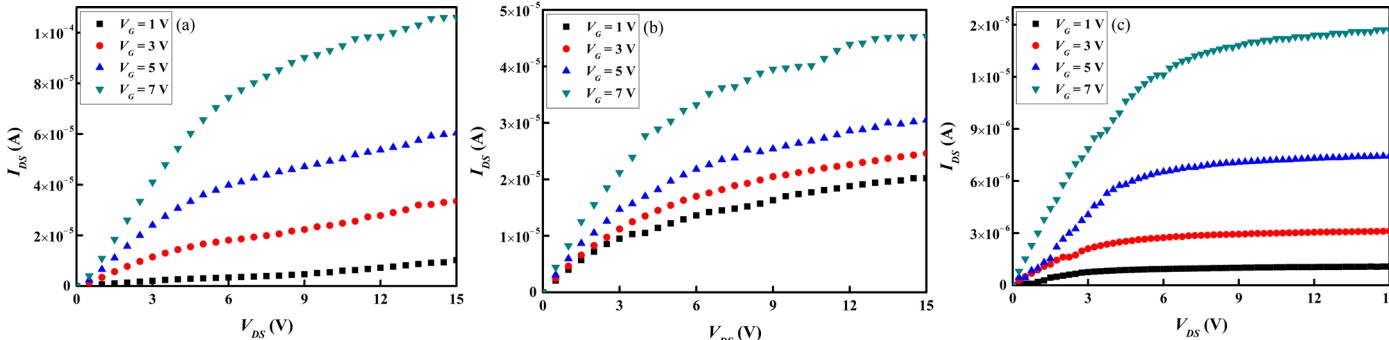

**Figure 14.** Output characteristics of TFT devices containing (**a**) *a*-InGaZn$_2$O$_5$, (**b**) *a*-InGaZnO$_4$, and (**c**) *a*-InGa$_2$ZnO$_{5.5}$ active channel layers subjected to post-annealing at 300 °C for 1 h in an ambient atmosphere.

InGaZnO$_4$ is the most popular type of active channel layer for *a*-IGZO TFT devices, and the transfer characteristics of such devices reported by previous studies are listed in Table 7. Compared with the TFT performance presented in Table 7, the *a*-InGaZnO$_4$ TFT device in this study should have achieved a satisfactory $\mu_{FE}$ property, whereas a further refinement in device structure is required to improve the $I_{on}/I_{off}$ value. As to the TFT devices containing other types of *a*-IGZO layers, Lee et al. [1] prepared an *a*-In$_2$Ga$_2$ZnO$_7$ layer for the TFT devices and obtained an $\mu_{FE}$ of 4.2 ± 0.4 cm$^2 \cdot$V$^{-1} \cdot$sec$^{-1}$ and an *SS* of 0.96 ± 0.10 V·decade$^{-1}$. This is inferior to the *a*-InGaZnO$_4$ TFT performance as listed in Table 7, implying the high Ga content of the *a*-IGZO channel layer might deteriorate the $\mu_{FE}$ and *SS* properties of the TFT devices. Shin et al. [4] deposited the *a*-IGZO layers at various oxygen ratios and their composition analysis found the Ga content of the *a*-IGZO layers increases with the oxygen ratio. Due to the tight attraction of the Ga$^{3+}$ ions to the oxygen ions, which consequently reduces the number of charge carriers generated by the oxygen vacancies, Shin et al. observed a decrement in the $\mu_{sat}$ and *SS* values of the TFT devices when the Ga content of the *a*-IGZO layer increases [4]. As illustrated by the transfer characteristics of *a*-IGZO TFT devices listed in Table 6, similar effects of Ga content on TFT performance were also observed in this study. The incorporation of Ga in *a*-IGZO channel layers might avoid the TFT devices in normally-on state; however, *a*-IGZO channel layers with appropriate Ga contents are essential to accomplish TFT devices with satisfactory electrical properties.

**Table 7.** A comparison of transfer characteristics of *a*-IGZO TFT devices containing an InGaZnO$_4$ active channel layer.

| $V_{th}$ (V) | $\mu_{FE}$ (cm$^2 \cdot$V$^{-1} \cdot$sec$^{-1}$) | *SS* (V·Decade$^{-1}$) | $I_{on}/I_{off}$ | Annealing Temp. (°C) | Ref. |
|---|---|---|---|---|---|
| −6.5 | 16 | 0.45 | | 250 | [3] |
| 13.9 | 2.6 | 0.93 | $1.1 \times 10^7$ | | [4] |
| 1.6 | 8.3 | | ~$10^3$ | | [5] |
| 0.57 | 14.7 | 0.45 | ~$10^8$ | 300 | [11] |
| 1.3 | 10 | 0.3~1.5 | ~$10^5$ | 500 | [16] |
| | 12 | | | | [18] |
| 5.5~13.6 | 8.4 | 0.62~0.9 | ~$10^8$ | | [21] |
| −0.85 | 8.46 | 2.31 | $2.9 \times 10^4$ | 300 | This study |

## 4. Conclusions

This study successfully prepares three types of IGZO sputtering targets with various Ga contents via the sintering of In$_2$O$_3$, Ga$_2$O$_3$, and ZnO raw powder mixtures at 900 °C to 1300 °C for 6 or 8 h. XRD analysis revealed that the InGaZn$_2$O$_5$ and InGaZnO$_4$ targets are of single-phase structure whereas the InGa$_2$ZnO$_{5.5}$ target is composed of

two phases, i.e., $Ga_2ZnO_4$ and $In_2O_3$. The IGZO layers were then deposited using self-prepared targets and incorporated in TFT devices to serve as active channel layers. XRD analysis indicated that all the IGZO layers are amorphous and without the presence of oxide compounds, and UV-NIR analysis found that their visible-light transmittances are above 80%. The results of Hall effect measurement found that the $N$ and $\mu_{Hall}$ decrease whereas the $\rho$ increases, with the increase of Ga content in the $a$-IGZO layers. This implied that the inhibition of $Ga^{3+}$ ions to the migration of charge carriers is the main factor affecting the electrical properties of $a$-IGZO, which is consistent with the results reported by Hosono [18]. The Ga contents in the $a$-IGZO channel layers similarly affected the transfer characteristics of the TFT devices prepared in this study. Post-annealing at 300 °C for 1 h in an ambient atmosphere might effectively improve the transfer characteristics of TFT devices. The best performance was observed in the $a$-InGaZnO$_4$ TFT sample subjected to post-annealing at 300 °C with $V_{th} = -0.85$ V, $\mu_{FE} = 8.46$ cm$^2$·V$^{-1}$·sec$^{-1}$, $SS = 2.31$ V·decade$^{-1}$, and $I_{on}/I_{off} = 2.9 \times 10^4$.

**Author Contributions:** Conceptualization, T.-E.H.; methodology, T.-E.H., T.-C.T. and J.-S.W.; software, T.-C.T.; data curation, J.-S.W.; writing—original draft preparation, T.-E.H. and T.-C.T.; writing—review and editing, T.-E.H., T.-C.T., J.-S.W. and H.-J.W. All authors have read and agreed to the published version of the manuscript.

**Funding:** This study was supported by the Ministry of Science and Technology (MOST), Taiwan, R.O.C., under the contract No. MOST 109-2221-E-009-053.

**Institutional Review Board Statement:** Not applicable.

**Informed Consent Statement:** Not applicable.

**Data Availability Statement:** Not applicable.

**Acknowledgments:** The authors thank the Nano Facility Center (NFC) of National Yang Ming Chiao Tung University for supplying the RF magnetron sputtering system and the PECVD systems.

**Conflicts of Interest:** The authors declare no conflict of interest.

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
