# Peer review of "The Fabrication of Indium–Gallium–Zinc Oxide Sputtering Targets with Various Gallium Contents and Their Applications to Top-Gate Thin-Film Transistors"

_coatings, doi:10.3390/coatings12081217_

Round 1

Reviewer 1 Report

This paper is systematically organized from basic research on synthesizing targets to device characteristic analysis. Therefore, I think it is appropriate to be published in a coating journal. However, I think the following minor corrections are necessary.

1. It is recommended to increase the resolution of the graphs in the paper.

2. In the experimental method of making IGZO targets on page 2, 90% of the target relative density is considered to be very low. Why is the relative density of the target low in this study and is there no effect on the electrical characteristics of the TFT device?

3. It is necessary to compare the electrical characteristics obtained through various measurements with the data of references.

Author Response

Response to the Reviewer 1 Comments

Manuscript ID: Coatings-1814811

Comment 1: It is recommended to increase the resolution of the graphs in the paper.

Response 1: Thanks for this comment. We have improved the resolutions of Figures 2, 3, 4, 5, 8, 9, 10, 11, 12 and 13. Please check the revised manuscript for reversion.

Comment 2: In the experimental method of making IGZO targets on page 2, 90% of the target relative density is considered to be very low. Why is the relative density of the target low in this study and is there no effect on the electrical characteristics of the TFT device?

Response 2: As described in line 76, page 2 of revised manuscript, we fabricated IGZO sputtering targets via the pressure-less sintering process in our study. Although the relative density of targets was relatively low as 90%, our SEM observation indicated the IGZO films with good quality can be obtained by using the self-prepared targets. Certainly the crystalline defects might present in IGZO films, it is however common to all thin films prepared by sputtering method. Our previous study (C.-C. Lo, et al. Ceramics International, 38(2012), p.3977) have demonstrated that the IGZO-TFT devices with satisfactory performance can be achieved using the target prepared by the method of this study and, as a matter of fact, similar results were also obtained in this study. We hence believe that the sputtering targets with the density of 90% has no dramatic effect on the electrical characteristics of the TFT devices.

Comment 3: It is necessary to compare the electrical characteristics obtained through various measurements with the data of references.

Response 3: Thanks for this comment. We thought of comparing the electrical performance of TFT devices when writing this article; however, literature survey indicated that most of the TFT studies adopting a-InGaZnO4 as the active channel layer and we are likely the first study investigating TFT devices containing a-IGZO layers with various Ga contents. Moreover, it is somewhat difficult to write the TFT performance comparison since the device structures, fabrication methods and annealing treatments vary among researches. We nevertheless tried our best to write the TFT performance comparison as follows. The statements in below can be found in page 11, lines 283-304 of revised manuscript.

InGaZnO4 is the most popular type of active channel layer for a-IGZO TFT devices and the transfer characteristics of such devices reported by previous studies are listed in Table 7. Compared with the TFT performance presented in Table 7, the a-InGaZnO4 TFT device of this study should have achieved a satisfactory mFE property whereas a further refinement of device structure is required in order to improve the Ion/Ioff value. As to the TFT devices containing other types of a-IGZO layers, Lee et al. [1] prepared a-In2Ga2ZnO7 layer for the TFT devices and obtained mFE of 4.2±0.4 cm2×V-1×sec-1 and SS of 0.96±0.10 V×decade-1. This is inferior to the a-InGaZnO4 TFT performance as listed in Table 7, implying the high Ga content of a-IGZO channel layer might deteriorate the mFE and SS properties of TFT devices. Shin et al. [4] deposited the a-IGZO layers at various oxygen ratios and the composition analysis found the Ga content of a-IGZO layers increases with the oxygen ratio. Due to the tight attraction of Ga3+ ions to the oxygen ions which consequently reduces the number of charge carriers generated by the oxygen vacancies, Shin et al. observed a decrement of msat and SS values of TFT devices when the Ga content of a-IGZO layer is increased [4]. As illustrated by the transfer characteristics of a-IGZO TFT devices listed in Table 6, similar effects of Ga content on the TFT performance were also observed by this study. Incorporation of Ga in a-IGZO channel layers might avoid the TFT devices in normally-on state; however, a-IGZO channel layers with appropriate Ga contents are essential to accomplish TFT devices with optimal electrical properties.

Table 7. A comparison of transfer characteristics of a-IGZO TFT devices containing InGaZnO4 active channel layer.

Vth(V)

mFE(cm2×V-1×sec-1)

SS(V×decade-1)

Ion/Ioff

Annealing Temp.(°C)

Ref.

-6.5

16

0.45

---

250

[3]

13.9

2.6

0.93

1.1´107

---

[4]

1.6

8.3

---

~103

---

[5]

0.57

14.7

0.45

~108

300

[11]

1.3

10

0.3~1.5

~105

500

[15]

---

12

---

---

---

[16]

5.5~13.6

8.4

0.62~0.9

~108

---

[19]

-0.85

8.46

2.31

2.9´104

300

This study

Reviewer 2 Report

The manuscript is on time in the field of TFTs development. New experimental results are presented. 

Nevertheless I should like to make some few critical comments. 

The introduction should be a bit corrected, where well recognized parameters of TFTs made with IGZO should be mentioned.  Authors mention  that they use implantation, various Ga  contents were implanted in TFT devices… - others meaning of the word “implantation” is used in semiconductor device technology, so the word should be actually changed. 

Fig. 4 presents transmittance of the 25 nm thick films, which are generally well transparent. It cannot be recognize difference in transmittance for such thin films, so these figures are not representative. Thicker films are more adequate to find correlation with atomic compositions of the films.

It is not clear, if transmittance is measured with dielectric films on the top for the 300°C-annealed a-IGZO layers. 

A post-annealing at  300°C for 1 h in ambient atmosphere was also performed for part of the TFT samples so as to explore the influence of heat treatment on device performance. - this is not discussed in the paper. 

Figure 6. SEM micrographs and EDS element mappings – crucial information about Oxygen content is omitted, so these figures do not  inform readers about composition , except planar uniformity of elements. 

Lines 233 – 236 describe general conclusion of annealing influence onto better TFT performance, but here is no discussion about important interfaces, or which physical processes are here possible to improve quality of the interface.  

Line  286 Conclusions about resistivity increases with the increase of Ga  content in a-IGZO layers - is not adequate. Here are compared 3 layers with different Ga content, but it is not known chemical compound. Here in the layers can be present mixture of oxides compounds, which determine more or less film resistivity.

Discussion should be more deeply provided to improve paper quality. 

Author Response

Response to the Reviewer 2 Comments

Manuscript ID: Coatings-1814811

Comment 1: The introduction should be a bit corrected, where well recognized parameters of TFTs made with IGZO should be mentioned. Authors mention that they use implantation, various Ga contents were implanted in TFT devices…- others meaning of the word “implantation” is used in semiconductor device technology, so the word should be actually changed.

Response 1: Thanks for this comment. We modified the statements in revised manuscript as follows:

  • Page 2, lines 58-62: “Afterward, the TFT devices containing IGZO active channel layers with various Ga contents were prepared and their transfer characteristics including threshold voltage (Vth), field-effect mobility (mFE), subthreshold swing (SS) and on/off current ratio (Ion/Ioff) were investigated. The effects of post-annealing on device performance were also examined and discussed as follows.
  • Page 11, lines 310-311: “The IGZO layers were then deposited using the self-prepared targets and incorporated in TFT devices to serve as the active channel layers.”

Comment 2: Fig. 4 presents transmittance of the 25 nm thick films, which are generally well transparent. It cannot be recognize difference in transmittance for such thin films, so these figures are not representative. Thicker films are more adequate to find correlation with atomic compositions of the films.

Response 2: Thanks for this comment. In this study, we measured the transmittance of IGZO layers with the thickness of 25 nm simply because the IGZO layers of 25-nm thick serve as the active channel layers of our TFT devices. The data of Fig. 4 might not be representative and we certainly could measure the transmittance of IGZO layers of various thicknesses, it however does not mean too much for subsequent electrical characterizations. Moreover, this is not a study of the optical properties of IGZO layers so that we did not measure the transmittance of thicker IGZO layers.

Comment 3: It is not clear, if transmittance is measured with dielectric films on the top for the 300°C-annealed a-IGZO layers.

Response 3: Thanks for this comment. The IGZO layers deposited on glass substrates are the samples for transmittance measurement. Hence, there is no dielectric films on the top of IGZO layers. The sample type for transmittance measurement has been added in page 3, lines117-119 of revised manuscript.

Comment 4: A post-annealing at 300°C for 1 h in ambient atmosphere was also performed for part of the TFT samples so as to explore the influence of heat treatment on device performance- this is not discussed in the paper.

Response 4: We have discussed the performance of TFT devices subjected to 300°C annealing. Please consult page 10, lines 265-274 of revised manuscript.

Comment 5: Figure 6. SEM micrographs and EDS element mappings– crucial information about oxygen content is omitted, so these figures do not inform readers about composition, except planar uniformity of elements.

Response 5: Thanks for this comment. First, we would like to point out that EDS mapping is qualitative that its results may not be proper for composition analysis. Secondly, our XRD analysis of IGZO targets indicated both InGaZn2O5 and InGaZnO4 targets are single-phase structures whereas the InGa2ZnO5.5 target is comprised of two phases, i.e., Ga2ZnO4 and In2O3. On the other hand, the IGZO layers deposited by using these targets were all amorphous and 300°C/1 h annealing did not alter the amorphism. So we adopted SEM/EDS to examine the In, Ga and Zn elements in IGZO layers, in particular in InGa2ZnO5.5 layer, are either randomly distributed or form the aggregates of oxide phases as observed by XRD. You can see that all elements are uniformly distributed and, hence, IGZO layers are amorphous without the presence of phase compound mixture. We believed that the EDS mapping of In, Ga and Zn elements should be sufficient to illustrate such results and hence omitted the EDS mapping of O element.

Comment 6: Lines 233 – 236 describe general conclusion of annealing influence onto better TFT performance, but here is no discussion about important interfaces, or which physical processes are here possible to improve quality of the interface.

Response 6: Thanks for this comment and we agree with your comment that the interface effects should be discussed in more obvious manner. In our previous study of a-IGZO TFT (C.-C. Lo, et al. Ceramics International, 38(2012), p.3977), the capacitance-voltage (C-V) analysis was performed and it was found that the post annealing at 300°C for 1 h is able to reduce the trap density (Dit) at a-IGZO/SiO2 interface for about one order of magnitude. This explains why we perform the 300°C/1 h annealing for TFT devices to improve their performance. Relevant discussion can be found in page 8, line 233-237 of revised manuscript.

Comment 7: Line 286 Conclusions about resistivity increases with the increase of Ga content in a-IGZO layers- is not adequate. Here are compared 3 layers with different Ga content, but it is not known chemical compound. Here in the layers can be present mixture of oxides compounds, which determine more or less film resistivity.

Response 7: Thanks for this comment. But we would like to point out that our XRD and SEM/EDS analyses have clearly shown that all IGZO layers are amorphous without the presence of phase compound and, hence, there should have no mixture of oxides compounds that affect the film resistivity. As a result, it is reasonable to infer that the Ga content of a-IGZO layers is the main factor relating to the film resistivity. We have rewritten the statement in Conclusions to emphasize the absence of oxide compound mixture in a-IGZO layers. Please consult page 11, line 311-313 of revised manuscript.
